## [Decision Letter · Decision Letter 0]

22 Sep 2025

Dear Dr. sun,

Thank you for submitting your manuscript to PLOS ONE. After careful consideration, we feel that it has merit but does not fully meet PLOS ONE’s publication criteria as it currently stands. Therefore, we invite you to submit a revised version of the manuscript that addresses the points raised during the review process.

We look forward to receiving your revised manuscript.

Kind regards,

Wislei Riuper Osório

Academic Editor

PLOS ONE

Journal Requirements:

“This work was support by 111 project of China [D21017]; Shandong Provincial Key Research and Development Program (International Science and Technology Cooperation)[No.2024KJHZ002]; Natural Science Foundation of Shandong Province [No. ZR2023ME156]; Double Hundred Plan" Talent Program of Shandong Province [WSR2023055];National Natural Science Foundation of China (52405492);Qingdao Natural Science Foundation [24-4-4-zrjj-68-jch].”

3. We note that your Data Availability Statement is currently as follows:

“All relevant data are within the manuscript and its Supporting Information files.”

5. We note you have included a table to which you do not refer in the text of your manuscript. Please ensure that you refer to Table 5 in your text; if accepted, production will need this reference to link the reader to the Table.

**Additional Editor Comments:**

Based on the Reviewer’s comments and suggestions, it is suggested that a MAJOR REVISION be provided. For this, all rebuttals should be detailed and solved

Reviewers' comments:

Reviewer's Responses to Questions

**Comments to the Author**

1. Is the manuscript technically sound, and do the data support the conclusions?

Reviewer #1: Partly

2. Has the statistical analysis been performed appropriately and rigorously?

Reviewer #1: No

3. Have the authors made all data underlying the findings in their manuscript fully available?

Reviewer #1: Yes

4. Is the manuscript presented in an intelligible fashion and written in standard English?

Reviewer #1: Yes

Reviewer #1: Clarity of Objectives

The introduction is comprehensive but overly long. The novelty of combining Cu and Nb should be emphasized earlier, with a clearer statement of the research gap that differentiates this study from prior work on Ti, Mo, CeO₂, and other elements.

Consistency of Microhardness Values

There is a discrepancy between the values reported in Section 3.3 (C5 ~696 HV) and the Conclusions (769 HV). The authors should verify and harmonize all numerical values throughout the manuscript.

Statistical Analysis

While the methods mention repeating electrochemical tests at least three times, the figures (hardness, friction, EIS, polarization) do not show error bars or deviations. Including statistical data would increase the reliability of the results.

Figures and Captions

Several figure legends are descriptive rather than interpretative (e.g., “Microstructure of coatings C1–C6”). To improve scientific communication, captions should highlight the main findings (e.g., “Transition from equiaxed to cellular microstructure with Nb addition”).

Wear Mechanism Evidence

The discussion identifies adhesive and abrasive wear, but this is not fully supported by quantitative analysis. Could the authors provide wear track depth, wear volume, or other quantitative evidence to substantiate the proposed mechanisms?

Corrosion Mechanism

The manuscript provides a strong narrative on the role of Cu and Nb in corrosion resistance. However, a schematic diagram showing the formation and stabilization of the passive film would enhance clarity and impact.

References

The reference list is up to date, but some entries appear truncated or inconsistently formatted. A careful review of the reference style and completeness is necessary to meet journal requirements.

Language and Style

The manuscript is readable, but several sections contain long, repetitive sentences. A thorough language polishing would improve fluency and conciseness.

**Do you want your identity to be public for this peer review?** For information about this choice, including consent withdrawal, please see our Privacy Policy

Reviewer #1: **Yes:** Prof. Dr. Yuri Alexandre Meyer

---

## [Author Response · Author response to Decision Letter 1]

1 Dec 2025

Response to Reviewers

Manuscript ID: PONE-D-25-46975

Title: Study on the Microstructure and Properties of Laser-Cladded AlCoCrFeNiCu₀.₅₋ₓNbₓ High-Entropy Alloy Coatings

We sincerely appreciate the Academic Editor and Reviewer #1 for taking the time to review our manuscript. We are grateful for the constructive and valuable comments, which have greatly helped us improve the quality, clarity, and rigor of the work. We have carefully revised the manuscript according to all suggestions. Below, we provide detailed point-by-point responses.

All modifications in the revised manuscript are highlighted using Track Changes. In the article, the parts marked in blue font are deleted, and the parts marked in red font are newly added

Academic Editor Comments and Author Responses

1.Manuscript Formatting and Style Compliance

Response：

We have revised the manuscript to conform fully to the PLOS ONE formatting guidelines. File naming, figure formatting, table labeling, font consistency, and section organization have been corrected according to the template instructions.

2. Funding Statement Correction

Response:

The previously included funding information in the Acknowledgments section has been removed and the Funding Statement has been updated.

The corrected Funding Statement is as follows:

Funding Statement:

This work was supported by the 111 Project of China [D21017]; Shandong Provincial Key Research and Development Program (International Science and Technology Cooperation) [2024KJHZ002]; Natural Science Foundation of Shandong Province [ZR2023ME156]; "Double Hundred Plan" Talent Program of Shandong Province [WSR2023055]; National Natural Science Foundation of China [52405492]; and Qingdao Natural Science Foundation [24-4-4-zrjj-68-jch].

The Acknowledgments section has been updated accordingly.

3. Data Availability Confirmation

Response:

All raw data necessary to replicate the study results, including numerical values behind plotted graphs, hardness and friction data, have now been provided as Supporting Information files (S1 Dataset).

We have updated the Data Availability Statement as follows:

Data Availability:

All relevant data are included within the manuscript and its Supporting Information files (S1 Dataset).

4. ORCID Validation

Response:

The corresponding author has completed ORCID registration and validation through Editorial Manager.

5. We note you have included a table to which you do not refer in the text of your manuscript. Please ensure that you refer to Table 5 in your text; if accepted, production will need this reference to link the reader to the Table.

Response:

A clear in-text citation to Table 5 has been added in Section3.5 Corrosion Friction Performance of the manuscript.

6. Citation Recommendations

Response:

We reviewed all suggested literature and incorporated citations in cases where the referenced works were relevant and supportive to the context. Citations were not added where relevance could not be established to avoid unnecessary or forced referencing.

Reviewer Comments and Author Responses

1. Clarity of Objectives / Novelty Not Highlighted Early

Reviewer Comment:

The introduction is comprehensive but overly long. The novelty of combining Cu and Nb should be emphasized earlier, with a clearer statement of the research gap.

Response:

Thank you very much for this important suggestion. We have shortened the Introduction and moved the novelty statement forward, emphasizing the unique role of the Cu+Nb dual-alloying strategy and how it differs from previous approaches using single strengthening elements such as Ti, Mo, and CeO₂.

Revision Made:

Section 1 (Introduction), Delete some content from paragraphs 4, 5 and 7, Paragraph 8 rewritten.

Revised Sentence Added:

In this study, AlFeCoCrNiCu₀.₅₋ₓNbₓ (x = 0, 0.1, 0.2, 0.3, 0.4, 0.5) high-entropy alloy (HEA) coatings with varied Cu and Nb contents were fabricated by laser cladding. The synergistic effects of Cu and Nb on the coatings’ microstructure, phase constitution, hardness, friction and wear behavior, and corrosion resistance were systematically evaluated. Guided by a “structure–interface–environment” co-optimization framework, Cu was employed to stabilize the solid-solution matrix, densify the microstructure, and improve heat/mass-transport pathways, while Nb was utilized to regulate phase architecture and reinforce the robustness of the surface passive film, thereby balancing the dual requirements of wear resistance and corrosion protection. This strategy differs from conventional single-element approaches and highlights a key novelty: by tuning the Cu/Nb ratio, a simultaneous compromise between “toughness–hardness” and “densification–passivation” is achieved. The results elucidate how elemental interactions govern performance optimization, providing theoretical guidance and a practical reference for the deployment of AlFeCoCrNiCu₀.₅₋ₓNbₓ coatings in applications demanding enhanced anti-wear and anti-corrosion capabilities.

2. Consistency of Microhardness Values

Reviewer Comment:

There is a discrepancy between Section 3.3 (C5 ~696 HV) and the Conclusions (769 HV).

Response:

We appreciate the reviewer for pointing this out. We carefully checked the original hardness data and confirmed that the correct microhardness of C5 is 696 HV, not 769 HV. The error in the conclusion section has been corrected.

Revision Made:

Section 4 (conclusion) Paragraph 3

Revised Sentence Added:

The coating achieves its highest hardness (696.11 HV) at x = 0.4, representing a 228.4% increase over the substrate.

3. Statistical Analysis and Error Bars

Reviewer Comment:

The figures do not show error bars. Statistical analysis should be included.

Response:

We fully agree. We have now included error bars representing standard deviations for:

Hardness results ,Friction coefficient (COF) and Wear rate

Revision Made:

Figures 6 and 9 updated to include error bars.

4. Figure Captions Need to Highlight Main Findings

Reviewer Comment:

Captions are descriptive rather than interpretative.

Response:

We have revised all figure captions to highlight the key scientific findings rather than merely labeling the figures.

Example Change:

Fig 2. XRD pattern of AlFeCoCrNiCu0.5-xNbx HEAs coating(a)20-90°; (b) Enlarged view at 44-45°.

Fig 3. Transition from equiaxed to cellular microstructure with Nb addition (a) C1; (b) C2 (c) C3; (d) C4; (e) C5; (f) C6.

Fig 8. Three-dimensional wear track morphologies and corresponding cross-sectional profiles of the coatings after wear (a) C1; (b) C2 (c) C3; (d) C4; (e) C5; (f) C6.

Fig 10. Morphology of AlFeCoCrNiCu0.5-xNbx HEAs coating after wear (a) C1; (b) C2 (c) C3; (d) C4; (e) C5; (f) C6.

Fig 15. Schematic of passive-film and microstructure regulation by Cu and Nb in AlFeCoCrNiCu0.5₋ₓNbₓ laser-cladded coatings (a)Optimal (b)Non-optimal.

5. Wear Mechanism Evidence Needs Quantitative Support

Reviewer Comment:

Provide wear track depth or wear volume.

Response:

We have now added:

3D confocal wear track morphologies, Cross-sectional depth profiles, Wear volume estimation and specific wear rate comparison.These are included as new Figures 8 and 9.

This additional quantitative evidence strongly supports the identification of adhesive vs. abrasive wear mechanisms.

Newly added content:

Fig 8 shows the three-dimensional wear track morphologies of coatings C1–C6 obtained by laser confocal microscopy, together with the corresponding cross-sectional profiles. The quantitative measurements further substantiate the above trends. For C1, the wear track width and depth are about 635.2 μm and 10.83 μm, respectively, indicating a relatively large material removal volume, which is consistent with its higher specific wear rate (9.72 × 10⁻⁵ mm³·N⁻¹·m⁻¹). The C2 coating exhibits a narrower and slightly shallower wear scar (471.9 μm, 9.45 μm), corresponding to a reduced wear rate of 6.25 × 10⁻⁵ mm³·N⁻¹·m⁻¹. The C3 coating shows the narrowest track width (296.5 μm) with a moderate depth (13.23 μm), resulting in the smallest effective cross-sectional area and thus the lowest wear rate (2.70 × 10⁻⁵ mm³·N⁻¹·m⁻¹) among all coatings. This observation is in good agreement with its lowest COF, confirming that the optimized Cu/Nb ratio in C3 effectively suppresses both friction and wear.

In contrast, the C4 coating presents the widest and deepest wear track (707.9 μm, 16.27 μm), implying a significantly larger worn volume, which corresponds well to its highest wear rate (1.60 × 10⁻⁴ mm³·N⁻¹·m⁻¹) and highest COF. The C5 coating exhibits an intermediate wear scar (368.1 μm, 8.96 μm) and a moderate wear rate (4.58 × 10⁻⁵ mm³·N⁻¹·m⁻¹), while the C6 coating shows a relatively wide and deep track (469.7 μm, 10.41 μm) with an increased wear rate of 6.77 × 10⁻⁵ mm³·N⁻¹·m⁻¹. As illustrated in Fig 9, the variation of specific wear rate follows a similar tendency to that of the steady-state COF: C3 exhibits the lowest COF and wear rate, C4 shows the highest values for both, and the other compositions lie in between. This consistent correlation between friction behavior, wear rate, and 3D wear morphology further verifies the reliability of the tribological evaluation.

6. Corrosion Mechanism Would Benefit from a Schematic Diagram

Reviewer Comment:

A passive film formation schematic should be added.

Response:

We agree and have added a new schematic (now Fig. 15) that visualizes and the schematic diagram was analyzed:

Cu effect on FCC stabilization, Nb-promoted Laves phase dispersion,Formation and repair of Cr₂O₃–Nb₂O₅–Al₂O₃ passive film,This significantly improves clarity of the corrosion mechanism discussion.

Revision Made:

Section 3.8 (Corrosion mechanism of AlCoCrFeNiCu0.5-xNbₓ coatings) Paragraphs 2,3 and 5

Newly added content:

As schematically summarized in Fig 15 (a), moderate Cu works in concert with Cr, Nb and Al to build a dense composite film (Cr₂O₃ backbone, Nb₂O₅ defect suppression/re-passivation, and Al₂O₃ stability), which is reflected by the upward arrows indicating Ecorr↑, Icorr↓, and Rp↑. In contrast, Fig 15(b) illustrates two non-optimal cases: (i) excess Cu favors a Cu₂O-containing, less cohesive film; and (ii) excess Nb yields continuous grain-boundary (GB) Laves (purple band), which facilitates intergranular attack and undermines passivation.

This behavior is consistent with Fig 15, where the balanced Cu–Nb condition produces a structurally coherent, self-healing film (a), whereas Cu-rich or Nb-rich conditions deteriorate film integrity and electrochemical metrics (b).

Accordingly, Fig 15 (a) corresponds to the C3 condition: an FCC matrix + finely dispersed Laves “pinning” particles beneath a Cr₂O₃–Nb₂O₅–Al₂O₃ film, delivering the best combination of highest Ecorr, lowest Icorr, and largest Rp among the investigated coatings; the right-hand panels depict the Cu-rich and Nb-rich departures from this optimal balance.

7. Reference Formatting Issues

Reviewer Comment:

Some entries appear truncated or inconsistent.

Response:

We have thoroughly checked and corrected all reference formatting to meet PLOS ONE reference style requirements.

8. Language and Style Polishing

Reviewer Comment:

Several sections contain long or repetitive sentences.

Response:

We have revised the manuscript to improve language clarity, remove redundancy, and streamline transitions. We also performed full professional proofreading to ensure grammatical correctness.

Closing Statement

We sincerely thank Academic Editor and Reviewer #1 for the thoughtful and encouraging comments. We believe these revisions have substantially improved the manuscript, and we hope it is now suitable for publication in PLOS ONE.

---

## [Editor Report · Decision Letter 1]

2 Dec 2025

Dear Dr. sun,

Thank you for submitting your manuscript to PLOS ONE. After careful consideration, we feel that it has merit but does not fully meet PLOS ONE’s publication criteria as it currently stands. Therefore, we invite you to submit a revised version of the manuscript that addresses the points raised during the review process.

We look forward to receiving your revised manuscript.

Kind regards,

Wislei Riuper Osório

Academic Editor

PLOS ONE

**Journal Requirements:**

**Additional Editor Comments:**

Although it is observed that a great number of improvements are included, there still remained some weaknesses to be solved before its final publication, as followed:

1. The abstract should be reworked in order to only “SIMPLE PRESENT” verbal tense be used.

2. All subsections of the section 2 should be revised and all dimensions and values accompanied with their corresponding error ranges. At least the equipment error should be considered.

3. In subsection 2.4, it is stated that a potential scan rate of 1mV/s is used. Considering this point, the followed sentences and its references should be cited/included.

“It is remarked that potential scan rate has an important role in order to minimize the effects of distortion in Tafel slopes and corrosion current density analyses, as previously reported [AA-CC]. However, based on these reports, it is experimentally observed that the adopted 1 mV/s has no deleterious effects on those Tafel extrapolations to determine the corrosion current densities of the examined samples.”

[AA] Duarte T, Meyer Y.A. Osório W.R. The Holes of Zn Phosphate and Hot Dip Galvanizing on Electrochemical Behaviors of Multicoatings on Steel Substrates. Metals 2022, 12(5): 863; https://doi.org/10.3390/met12050863

[BB] Zhang X.L., Jiang Zh.H., Yao Zh.P, Song Y., Wu Zh.D. Effects of scan rate on the potentiodynamic polarization curve obtained to determine the Tafel slopes and corrosion current density. Corrosion Science. 2009, 51: 581-587.

[CC] E. McCafferty. Validation of corrosion rates measured by Tafel extrapolation method. Corr. Scie 47 (2005) 3202-3215 .

4. Considering Fig. 2, it is obligatory that its corresponding JCPDS file numbers be included. This can be made into the main text and optionally also into the corresponding figure caption.

5. At Table 3, the hardness unit is equivocate. It should be replaced with “HV” instead “Hv”. Additionally, its corresponding error ranges should be depicted. This is a mandatory action.

6. Nyquist plots shown into Figs. 13 and 14 are depicted erroneously. This considering the Y and X axes. Based on the fact the Nyquist plots can depict depressed semi arcs, it is mandatory that both X and Y axes be in same scale. This is a commonly and conventional practice among the corrosionists and electrochemical researchers.

7. Tafel extrapolations should be depicted into Fig. 13(d) and Fig. 14(d). These will confirm those values reported. It seems that there exists some no agreeing with those reported. Please, revise it.

8. Another weakness point concerns to the CNLS simulation and corresponding equivalent circuit. Since EIS parameters are determined, at least, it is expected that an equivalent circuit be depicted and discussed. Also, CNLS simulations lines (fitting) is obligatory to be included into the revised version of the manuscript. To guide line the Authors concern to CNLS, i.e. complex non-linear least squares (CNLS) simulation, there are some reference to be consulted:

[DD] Y. A. Meyer, R.S. Bonatti, A.D. Bortolozo, W.R. Osório. Electrochemcial behavior and compressive strength of Al-Cu/xCu composites in NaCl solution. Journal of Solid State Electrochemistry, 25(2021) 1-15, https://doi.org/10.1007/s10008-020-04890-x

[EE] Bryan Hirschorn and A. Lasia (book: Electrochemical impedance spectroscopy and its applications, 2014).

[FF] B. Hirschorn, M. E. Orazem, B. Tribollet, V. Vivier, Isabelle Frateur and M. Musiani. Determination of effective capacitance and film thickness from constant-phase-element parameters. Electrochimica Acta, 55 (2010) 6218-6227 .

9. Another electrochemical weakness verified in the proposed manuscript is the values “Icorr” shown into Table 7. If really “Icorr” are determined, it seems that new sentences should be included to explain its calculation and limitation. However, it seems that Authors have tried to determine the experimental corrosion current density, i.e. “icorr”. This is calculated from those Tafel (anode and cathode branches from the potentiodynamic polarization curves), while “Icorr” means the corrosion rate, and it is differently determined. This is also obligatory revision.

---

## [Author Response · Author response to Decision Letter 2]

19 Jan 2026

We thank the Academic Editor and the reviewers for their constructive comments on our manuscript (PONE-D-25-46975R1), “Study on the Microstructure and Properties of Laser-Cladded AlCoCrFeNiCu₀.₅₋ₓNbₓ High-Entropy Alloy Coatings.” We have revised the manuscript thoroughly and addressed all editor/reviewer comments point-by-point.

In the revised submission, we provide: (1) a marked-up manuscript with Track Changes, (2) a clean revised manuscript, and (3) a separate “Response to Reviewers” document . Major updates include revising the Abstract tense, adding uncertainty/error ranges in Section 2, clarifying the polarization scan-rate rationale with appropriate citations, adding JCPDS/PDF numbers for XRD, correcting hardness units and adding error ranges, redrawing Nyquist plots with identical axis scaling, adding Tafel extrapolation lines, and including CNLS fitting curves and the equivalent circuit for EIS; we also corrected the notation and explanation of icorr.We hope the revised manuscript is now suitable for further consideration.

---

## [Editor Report · Decision Letter 2]

25 Jan 2026

Study on the Microstructure and Properties of Laser-Cladded AlCoCrFeNiCu0.5-xNbₓ High-Entropy Alloy Coatings

PONE-D-25-46975R2

Dear Dr. sun,

We’re pleased to inform you that your manuscript has been judged scientifically suitable for publication and will be formally accepted for publication once it meets all outstanding technical requirements.

Kind regards,

Wislei Riuper Osório

Academic Editor

PLOS One

Additional Editor Comments (optional):

Based on the revised version of the proposed manuscript, it is observed that all suggestions and modifications were provided. With this, it deserves its final publication.
---

## [Editor Report · Acceptance letter]

PONE-D-25-46975R2

PLOS One

Dear Dr. Sun,

I'm pleased to inform you that your manuscript has been deemed suitable for publication in PLOS One. Congratulations! Your manuscript is now being handed over to our production team.

Kind regards,

on behalf of

Dr. Wislei Riuper Osório

Academic Editor

PLOS One